# Variability of Human rDNA and Transcription Activity of the Ribosomal Genes

**DOI:** 10.3390/ijms232315195

**Published:** 2022-12-02

**Authors:** Nikola Chmúrčiaková, Evgeny Smirnov, Jaroslav Kurfürst, František Liška, Dušan Cmarko

**Affiliations:** 1Laboratory of Cell Biology, Institute of Biology and Medical Genetics, First Faculty of Medicine, Charles University and General University Hospital in Prague, 128 01 Prague, Czech Republic; 2Institute of Organic Chemistry and Biochemistry of the Czech Academy of Sciences, 160 00 Prague, Czech Republic

**Keywords:** rDNA, sequence variability, transcription, non-canonical DNA structures, micro-RNAs

## Abstract

Human ribosomal DNA is represented by hundreds of repeats in each cell. Every repeat consists of two parts: a 13 kb long 47S DNA with genes encoding 18S, 5.8S, and 28S RNAs of ribosomal particles, and a 30 kb long intergenic spacer (IGS). Remarkably, transcription does not take place in all the repeats. The transcriptionally silent genes are characterized by the epigenetic marks of the inactive chromatin, including DNA hypermethylation of the promoter and adjacent areas. However, it is still unknown what causes the differentiation of the genes into active and silent. In this study, we examine whether this differentiation is related to the nucleotide sequence of IGS. We isolated ribosomal DNA from the nucleoli of human-derived HT1080 cells, and separated methylated and non-methylated DNA by chromatin immunoprecipitation. Then, we used PCR to amplify a 2 kb long region upstream of the transcription start and sequenced the product. We found that six SNVs and a series of short deletions in a region of simple repeats correlated with the DNA methylation status. These data indicate that variability of IGS sequence may initiate silencing of the ribosomal genes. Our study also suggests a number of pathways to this silencing that involve micro-RNAs and/or non-canonical DNA structures.

## 1. Introduction

Human ribosomal DNA is represented by hundreds of repeats or units in each cell. Every repeat consists of two parts: a 13 kb long 47S DNA with genes encoding 18S, 5.8S, and 28S RNAs of ribosomal particles, and a 30 kb long intergenic spacer (IGS) [1,2,3]. The transcription of rDNA effected by RNA polymerase I (pol I) is very intensive; however, it does not take place in all the repeats. The transcriptionally active genes are characterized by the epigenetic marks of the active chromatin, including hypomethylation of the promoter and adjacent IGS areas [4,5,6,7]. A significant part in the regulation of the transcription activity is played by IGS. It has a complex structure and contains various functional or supposedly functional regions producing non-coding RNAs, which participate in pol I activity regulation, as well as stress reactions and other processes in the cell nucleus and cytoplasm [8,9]. Especially important for rDNA silencing is a 2 kb long regulatory region upstream of the transcription start site (URR). This region includes sequences producing two long non-coding RNAs (lncRNA), promoter RNA (pRNA), and promoter and pre-rRNA antisense RNA (PAPAS). The first of these molecules is transcribed by pol I from a region beginning at about −2 kb and stretching into the 47S DNA. The primary transcript is processed into a set of 150–300 nt functional molecules, which can inhibit pol I transcription initiation through the recruitment of the nucleolar remodeling complex (NoRC) [10,11]. PAPAS is produced in the anti-sense direction by RNA polymerase II (pol II) from a region covering the URR and a large part of the ribosomal genes; this lncRNA is actually represented by a heterogenous pool of 12–16 kb long transcripts arising from multiple start sites. PAPAS can inhibit rDNA transcription in different ways, mainly by recruiting the histone methyltransferase Suv4–20H2 to the rDNA promoter [12,13,14]. Apparently, both IGS and 47S DNA encode many other transcripts contributing to the regulation of pol I activity. They include various types of small RNAs, especially micro-RNAs (miRs) and RNAs produced by the transposable Alu elements [8]. Recently, it has been found that non-canonical structures, particularly abundant in the rDNA locus, such as R-loops, G-quadruplexes, intercalated motifs, and triplexes, may also be important factors in the silencing/activating of ribosomal genes [15,16,17,18,19]. E.g., pRNA and PAPAS were found to form triplexes in IGS, which seem to be an indispensible condition for the silencing effected by these lncRNAs [19].

However, although many factors engaged in the locus regulation have been discovered, the essential question remains unanswered: how are the rDNA repeats differentiated into active and silent? That is to say, why do some repeats pass into a state favorable for transcription, i.e., acquire the proper chromatin structure and mode of regulatory non-coding RNA expression, while the other repeats in the same cell remain silent?

The answer is probably suggested by the fact that the rDNA locus is highly variable, probably because of the multiplicity of the units, which causes instability [9]. In this study, we examine whether the differentiation of rDNA into active and silent units is related to the nucleotide sequence of IGS. We isolated ribosomal DNA from the nucleoli of human fibrosarcoma cells (HT1080), and separated methylated and non-methylated DNA by chromatin immunoprecipitation with antibodies against 5-methylcytosine. Then, we used PCR to amplify the regulatory region URR and sequenced the products to find the frequencies of all the significant variants. Our study reveals a correlation between the sequence variability in the region and its methylation status, indicating that certain single nucleotide substitutions and short deletions may play a key role in silencing ribosomal genes. The results of this work also suggest a number of pathways to this silencing that involve micro-RNAs and/or non-canonical DNA structures.

## 2. Results

### 2.1. General Characteristics of the Observed Variability in the HT1080 Cells

In order to enrich the rDNA content, the DNA extracted from the isolated nucleoli was separated into the fractions of predominantly methylated and predominantly non-methylated with subsequent PCR amplification of the 2 kb long regulatory region upstream of the transcription start (URR) (Figure 1). Then, both fractions were subjected to deep sequencing (see Methods), which revealed multiple variants determined in reference to the human rDNA GenBank sequence U13369.1 (https://www.ncbi.nlm.nih.gov/nuccore/U13369.1?report=graph&from=19514, accessed on 1 October 2022).

The results are presented in Table 1 and Table 2.

Among the variants that appeared with the frequency >5%, we found totally: 33 SNVs, 5 simple inserts and deletes (indels), and a number of variously sized (from 1 to 34 bp) deletions in two regions of simple repeats (CCCT)n: Q1 (41667–41686; *n* = 5) and Q2 (41842–41877; *n* = 9), both flanked by CT and GC upstream and downstream, respectively. One SNV and one insert (at 41006 and 41008) belonged to the Alu element; one SNV at 42945 is situated between the UCE and CPE of the rDNA promoter. In all, 21 variants occurred with low frequency (5–30%). In addition, 13 variants seem to be a specific feature of the studied cell line, since they appeared in 70–100% reads. Moreover, three variants were found in 30–70% reads and, accordingly, the respective positions may be called hot spots of variability.

### 2.2. Comparing Variants in the Methylated and Non-Methylated Pools

Most variants did not significantly correlate with the methylation status. Among SNVs, significant correlation was observed at only seven positions: highly variable SNV at 41574; and low variability SNV at 41574, 41675, 41679, 41680, 41831, and 42329. All these variants appeared more frequently in the methylated than in the non-methylated fractions of rDNA. 

Since the repeats with CCCT motif seem to be optimal for the formation of non-canonical DNA structures, especially G-quadruplexes, we assembled the deletions of various lengths in the regions Q1 and Q2 as separate groups of data (Table 2). Comparing the deletion frequency in the methylated and non-methylated fractions, we observed a significant difference in Q1, but not in Q2 (Table 2). In the latter, almost all the repeats had deletions independently of the methylation status.

Q1 and Q2 differed by the length of the deletions; however, in both regions, this length was expressed predominantly in fourfold numbers from 4 to 32 (Table 3). Each deletion in Q1 and Q2 typically had on its ends the motif CTCC.

Some SNVs, which correlated with methylation state (41675; 41679; 41680), also belonged to the Q1 region and reduced the number of repeats in it. It was, thus, the general tendency that the predominantly methylated fraction of rDNA was more variable, in particular, with less CCCT repeats than the predominantly non-methylated sequences.

### 2.3. Search for Micro-RNAs (miRs) Corresponding to the Discovered Variants

Looking for plausible pathways that may connect variations of the rDNA sequence with its transcription activity, we used the publicly available miRBase (https://www.mirbase.org/search.shtml, accessed on 1 October 2022), and searched for RNA and DNA sequences matching with areas containing the discovered variants. Two methods were employed for this purpose:(1)For each studied position n we selected (from the reference database) a series of 20 bp long sequences containing n and beginning at n − 19, n − 18,…, n. These 20 sequences were subsequently entered in the database. (2)For each variable position n, we selected one 49 bp long sequence situated between n − 24 and n + 24, which was searched in the database with BLASTN and SSEARCH. Both searches were run with default parameters and a species restriction set to “human”. Search results were sorted and evaluated based on e-value; however, match length and percent identity were taken into consideration as well.

Applying these procedures to the six discovered SNVs significantly correlating with rDNA methylation, we found three miRs matching the small areas that contained some of these variants (Table 4). One miR thus corresponded to three SNVs; the other two miRs each corresponded to one SNV.

Alignment of the miRs with human rDNA (U13369.1). SNVs are shown below in red; the corresponding bases of miRs are in bold. In hsa-miR-6739, the SNV (G>T) improves the alignment:
U13369.1CTGGGCCCG**C**GGCGGGCGTGGGG(42320-42344)
|||||||||**|**|||||||||||||
hsa-miR-6724CTGGGCCCG**C**GGCGGGCGTGGGG(1-23)


U13369.1GGAGGGAGGGA**G**GGA**GG**GAGGG(41664-41685)
.||.||..|||**|**|||**||**|...|
hsa-miR-4769AGACGGTAGGA**G**GGA**GG**GGATG(1-22)


U13369.1**T**GTTCTTTCTCCCTCCC(41831-41847)
**|**||||||||||..||||
hsa-miR-6739**G**GTTCTTTCTCTTTCCC(2-17)

Remarkably, hsa-miR-6724 is transcribed from the 92 bp long gene lodged within IGS, upstream of the promoter. SNV42329 appearing in about 30% of reads alters this gene. The other two miRs are transcribed from the sequences outside the rDNA. Hsa-miR-4769 partly matches to one of the (CCCT) boxes, namely Q1. This region includes three SNVs as well as various deletions. Hsa-miR-6739 matches to a region downstream of the position 41831; however, the discovered variant G>T (C>A in the anti-sense strand) moves this position into the miR consensus region.

## 3. Discussion

In this study, we established a correlation between the sequence variability and methylation status of the rDNA regulatory region. Since this status reflects the state of transcription activity [7,20], our data suggest that variability of the IGS sequence in individual rDNA units can initiate selective silencing/activation of the ribosomal genes (Table 1, Table 2 and Table 3). SNVs in six positions and deletions in the region Q1 may be the first links in the chain of events resulting in the inhibition of the transcription. Moreover, our data seem to indicate how the sequence variability can be translated into a crucial shift of methylation and transcription status. In this respect, the region Q1 is particularly remarkable; its variability included deletions as well as three SNVs (41675; 41679; 41680) (Table 1 and Table 2), all of which reduced the number of CCCT repeats. Importantly, these repeats are favorable for the formation of non-canonical DNA structures. The regular patterns with abundant cytosines often generate intercalated motifs (i-motifs), the quadruplex structures, which may participate in the regulation of gene expression [21,22,23]. Still more probable in the Q1 region is the formation of another quadruplex structure, G-quadruplex (G4), based on the complementary strand region (GGGA)_n_. Indeed, we find here a perfect agreement with the formula used for the prediction of G4 structures: G_a_N_1_G_b_N_2_G_c_N_3_G_d_N_4_, where G signifies guanine; a, b, c ≥ 3, N_1,2,3,4_ are intermediate loop sequences of 1 to 7 nt [24]. In our case, a, b, c = 3; N_i_ = 1. Recent studies have shown that G4 structures not only destabilize the function of the genome, but may also contribute to the transcription regulation in rDNA and other regions of the genome [25]. Accordingly, it seems probable that G4 in the intact Q1 region should inhibit production of the pRNA transcript. On the other hand, the deletions and SNVs reducing the number of CCCT repeats would prevent G4 formation and, thus, facilitate pRNA expression. Now, since pRNA causes the hypermethylation of rDNA and silencing of the ribosomal genes, the discovered variants in the Q1 region are apparently likely to contribute to, or even initiate, the transcription suppression. The same variability may similarly affect the expression of PAPAS, though perhaps only in a fraction of cells, for this lncRNA is usually transcribed in stress conditions [26].

The situation of two other SNVs (41831 and 42329) suggests additional pathways connecting the sequence variability and gene silencing. These variants are positioned in the areas well matched to certain miRs and there are reasons to believe that miR functions may be disrupted by single-nucleotide replacement [27,28]. SNV42329 is located within the gene of hsa-miR-6724 (42320–42342); and the putative targets of this miR seem to be related, for the most part, to cancer or cancer-linked pathways, such as MAPK signaling and ERBB signaling, which are closely connected to the rDNA methylation and transcription [29]. This indicates that SNV42329 may have an impact on the expression of ribosomal genes. SNV41831 makes the upstream end of an area matched to hsa-miR-6739. This miR is produced outside of the rDNA locus. Nevertheless, it still can have an impact on the transcription activity of rDNA, because R-loops, which are often generated in the locus by pol II and other factors, may function as regulators of gene expression or primers for replication [16,19,30,31]. The single DNA strands of the R-loops would enable miRs to bind to their consensus sequence in IGS. Thus, SNV41831 may also affect rDNA transcription through interaction with hsa-miR-6739, although the particulars of the process are still not clear. Finally, hsa-miR-4769 partly matches with the quadruplex favorable region Q1, where we find three SNVs: 41675; 41679, and 41680 (Table 3). However, the quadruplexes, especially G4, are mutually convertible with R-loops, and the two non-canonical states sometimes exist in a dynamic equilibrium [32,33]. This would enable hsa-miR-4769 to form a new R-loop or to stabilize an existing one by binding to the anti-sense strand of the region, when it is released from the quadruplex state. Another obstacle for the transcription of pRNA would thus be created. However, the three SNVs would prevent the interaction of the miR with Q1 and facilitate pRNA expression, which would result in silencing of the ribosomal genes.

Taken together, our results indicate that a number of SNVs and short deletions, which correlate with DNA methylation of the rDNA regulatory region, may contribute to or initiate silencing of the locus. All these variants, except for one SNV, proved to be related to the functional/potentially functional regions of rDNA, which allowed us to suggest certain hypothetical pathways connecting this silencing with the DNA variability: -by suppressing non-canonical structures generated in IGS and facilitating pRNA transcription; -by altering the structure of an miR produced in IGS;-by interfering with the activity of miRs transcribed outside the rDNA locus.

## 4. Materials and Methods

### 4.1. Cell Culture

The human fibrosarcoma cell line, HT1080, was obtained from American Type Culture Collection (Rockville, MD, USA). The cells were maintained in complete DMEM Dulbecco’s modified Eagle medium containing 10% FBS and 1% penicillin–streptomycin. All the chemicals were obtained from Merck Life Science (Prague, Czech Republic) unless otherwise stated. Cell cultures were maintained at 37 °C, in a humidified atmosphere of 5% CO_2_.

### 4.2. Extraction of the Nucleoli

Nucleoli of HT1080 cells were isolated as follows [34,35]. In short, the cells were homogenized in a hypotonic buffer (10 mM HEPES, 10 mM KCl, 1.5 mM MgCl_2_, and 0.5 mM dithiothreitol; pH 7.9) using a pre-cooled glass Dounce homogenizer. The nuclear pellet was resuspended in 0.25 M sucrose (plus 10 mM MgCl_2_) and cleared by centrifugation over a cushion of S2 solution (0.35 M sucrose and 0.5 mM MgCl_2_). The nuclei were resuspended in S2 solution and sonicated at high amplitude in 15 s intervals, followed by phase-contrast monitoring for nuclear disruption. The sonicated sample was layered onto a cushion of 0.88 M sucrose and 0.5 mM MgCl_2_, and the nucleolar pellet was recovered after centrifugation. All the steps were performed at 4 °C.

### 4.3. Methylated DNA Immunoprecipitation

The DNA was extracted from the nucleoli by a QIAmp DNA Mini Kit (QIAGEN, Hilden, Germany). Methylated DNA immunoprecipitation (MeDIP) was performed using the Methylated DNA Immunoprecipitation Kit (Abcam, Cambridge, UK) according to the manufacturer’s protocol. The total isolated DNA was sonicated (five pulses for 25 s at high amplitude with 45 s intervals between pulses while resting on ice) to yield DNA fragments of 500–1000 bp in length. We used 1 μg of fragmented DNA for the immunoprecipitation. After heat denaturation, the DNA was incubated for 2 h at room temperature with a 1 μL anti-5 mC antibody containing 1 μL of normal mouse IgG as the negative control. The supernatant was collected as a nonmethylated DNA fraction. The methylated DNA was released from the DNA/antibody complex using proteinase K at 65 ℃ for 1 h. Subsequently, the samples were purified in spin columns and eluted. Both DNA fractions were amplified by PCR with the primers listed in Table 5.

### 4.4. Sequencing and Bioinformatic Analysis

PCR products were sequenced on an Illumina MiSeq using the Nextera XT DNA Library Preparation Kit (Illumina, San Diego, CA, USA). The sequencing data were uploaded to the Galaxy web platform and we used the public server at usegalaxy.org for bioinformatic analysis [36]. Mapping was conducted by BWA-MEM. [37]. FreeBayes was used to identify sequence variants [38]. An IGV (integrated genomics viewer) was used for data visualization [39]. The numbers of reads overlapping Q1 and Q1 repeats (Figure 1, Table 2) were obtained with scripts employing samtools, bedtools, and other in-house scripts.

## Figures and Tables

**Figure 1 ijms-23-15195-f001:**
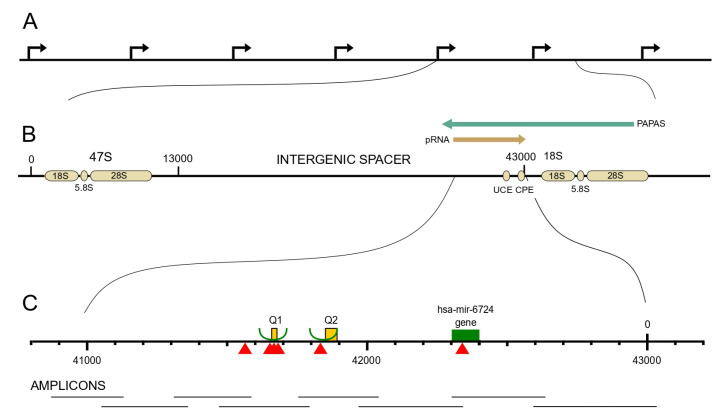
The structure of rDNA locus and the variants which appeared with different frequencies in the methylated and non-methylated DNA. (**A**) Cluster of repetitive rDNA units. (**B**) Details of one unit including genes encoding 18S, 5.8S, and 28S ribosomal RNAs; intergenic spacer (IGS); promoter with upstream control element (UCE) and core promotor element (CPE); promoter and pre-rRNA antisense (PAPAS) and promoter-associated RNA (pRNA). (**C**) The regulatory region upstream of the transcription start site (URR); the fragments amplified by PCR are shown below. Variants with significant difference of methylated and non-methylated state are indicated by red arrowheads. Yellow rectangles represent regions of simple repeats Q1 and Q2. The green rectangle corresponds to the gene encoding hsa-mir-6724. Two green semicircles indicate regions corresponding to miRs produced outside the rDNA locus.

**Table 1 ijms-23-15195-t001:** SNVs and simple indels in the methylated/non-methylated rDNA. Variants with significant difference of methylated and non-methylated state are framed in red. Hot spots of variation are in bold.

Position	Variant	Frequency		Coverage		Significance
		Methylated	Non-Methylated	Methylated	Non-Methylated	
41006	insG	0.99	0.99	41071	41396	*p* > 0.05
41008	G>C	0.99	0.99	20577	20767	*p* > 0.05
41074	T>G	0.16	0.17	19920	19203	*p* > 0.05
41134	A>G	0.19	0.17	4952	860	*p* > 0.05
41208	C>T	0.98	0.98	5317	920	*p* > 0.05
**41214**	**T>C**	**0.51**	**0.51**	**5395**	**935**	***p* > 0.05**
41310	delC	0.95	0.94	2031	325	*p* > 0.05
41479	G>C	0.25	0.23	2515	5844	*p* > 0.05
41560	delT	0.99	0.99	957	388	*p* > 0.05
**41574**	**C>T**	**0.48**	**0.69**	**1195**	**4467**	***p* < 0.001**
41594	delT	0.98	0.99	995	3990	*p* > 0.05
41675	C>A	0.06	0.00	358	832	*p* < 0.001
41679	C>A	0.07	0.00	289	489	*p* < 0.001
41680	C>A	0.11	0.00	220	279	*p* < 0.001
**41680**	**C>T**	**0.41**	**0.45**	**220**	**279**	***p* > 0.05**
41681	C>A	0.06	0.06	209	247	*p* > 0.05
41781	C>G	0.93	0.88	166	290	*p* > 0.05
41782	G>C	0.95	0.94	176	298	*p* > 0.05
41811	delC	0.99	0.98	252	411	*p* > 0.05
41831	G>T	0.05	0.01	236	411	*p* < 0.05
42045	A>C	0.05	0.04	5578	21939	*p* > 0.05
42255	C>G	0.99	0.99	3234	12700	*p* > 0.05
42256	G>C	0.99	0.99	3153	12313	*p* > 0.05
42301	A>G	0.05	0.06	1709	8932	*p* > 0.05
42329	C>A	0.33	0.29	1221	9237	*p* < 0.05
42338	T>G	0.05	0.05	1339	10181	*p* > 0.05
42387	A>G	0.07	0.08	1663	12473	*p* > 0.05
42400	C>G	0.97	0.95	1644	12349	*p* > 0.05
42459	T>C	0.10	0.11	11659	12256	*p* > 0.05
42773	G>A	0.05	0.06	11959	4163	*p* > 0.05
42828	T>C	0.06	0.06	15931	5564	*p* > 0.05
42945	T>C	0.10	0.10	13170	4952	*p* > 0.05

**Table 2 ijms-23-15195-t002:** Deletions in the methylated/non-methylated regions of simple repeats Q1 and Q2. The frequencies were counted from the total number of deletions. To assess the correlation of these deletions with methylation status, we selected only the reads that covered entire Q1 or Q2 regions plus 5 nucleotides upstream and downstream.

(CCCT)n Region	Frequency		Selected Reads Coverage		Significance
	Methylated	Non-Methylated	Methylated	Non-Methylated	
Q1 (41667–41686)	0.95	0.11	62	18	*p* < 0.0001
Q2 (41842–41877)	0.98	1.00	83	179	*p* > 0.05

**Table 3 ijms-23-15195-t003:** Length distribution of the deletions in the regions Q1 and Q2.

Region	Length	Frequency
Q1	12	0.47
	32	0.18
	28	0.20
	24	0.04
	16	0.07
	20	0.02
	1	0.02
	24	0.51
Q2	8	0.26
	28	0.18
	12	0.03
	7	0.01
	4	0.01
	1	0.01
Q1 + Q2	24	0.42
	8	0.22
	28	0.18
	12	0.11
	32	0.03
	7	0.01
	4	0.01
	1	0.01
	20	0.00

**Table 4 ijms-23-15195-t004:** Three miRs corresponding to the discovered SNVs.

miR Name	Orientation	Origin	SNV
hsa-miR-6724	Sense	IGS	42329, C>A
hsa-miR-4769	anti-sense	outside rDNA	41675, C>A 41679, C>A 41680, C>A
hsa-miR-6739	anti-sense	outside rDNA	41831, G>T

**Table 5 ijms-23-15195-t005:** Primers used for PCR amplification.

Position	Size of the Product (bp)	Forward Primer	Reverse Primer
40877–41134	257	GGCTTGGCTGATGTTTGTG	GCAGAGATACACGTTGTCGT
41051–41358	307	TCGTTTCCACGCGTTTAC	CACGAAGGAGAGGAAGAAAG
41311–41568	275	CCGAGCGTACGTAGTTATCTC	AAGACAGACGGGAAGGAAAG
41469–41790	321	CGTTCCCTGTGTTTCCTTCT	GCAGAATCGGTAGGCTCTTC
41748–42033	285	GTCTGTCTCTGCGTGGATTC	CGAAACCGTGAGTCGAGAAG
41953–42323	370	TTTGGGCACCGTTTGTGT	CAGAAGCGCAGCGACAG
42276–42607	332	TCTGGCGTGCAGGTTTATGT	GGACTCGCCAGAAAGGATCG
42589–43029	439	GATCCTTTCTGGCGAGTCC	ACAGGTCGCCAGAGGACAG

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
