# Peer review of "Variability of Human rDNA and Transcription Activity of the Ribosomal Genes"

_ijms, 2022, doi:10.3390/ijms232315195_

Round 1
Reviewer 1 Report
The authors have studied the variability of rDNA and related transcriptional activity of ribosomal genes. They found that six SNVs and some short deletions correlate with the methylation status. The variability of IGS may start silencing of the ribosomal genes.
The paper is clear and straightforward, and the results do support the conclusions. The only point which I would like to be further explained/amplified is the last part of the Discussion. here the authors propose three hypothetical pathways resulting in silencing of the genes in question.
Are there any possibility to prove/disprove these hypotheses? Are the authors working on it?
Last point, unrelated to the present paper but quoted as literature. PAPAS is produced by Pol II in antisense direction. This means Pol II must be present in the nucleolus. However, there is no trace of immunocytochemically detectable Pol II in the nucleolus in the literature. This looks as conflicting.
Author Response
Dear reviewer,
thank you for your comments. We revised the manuscript addressing all remarks of revisions. In particular, we provided the requested characterization of the discovered deletions.
Following are our answers:
>>>… the authors propose three hypothetical pathways resulting in silencing of the genes in question. Are there any possibility to prove/disprove these hypotheses? Are the authors working on it?
We hope that the present work will be the first in a series of studies in which the suggested hypotheses will be examined. In particular, we consider detecting the non-canonical structures on the preparations of spread DNA and chromatin; using RNAi on knockdown hsa-mir-6724; continuing the study of the discovered variants in the normal human cells and tumours.
>>>PAPAS is produced by Pol II in antisense direction. This means Pol II must be present in the nucleolus. However, there is no trace of immunocytochemically detectable Pol II in the nucleolus in the literature. This looks as conflicting.
Indeed, we ourselves used to claim in our previous works that all transcription in the nucleoli is effected by pol I. Now it seems that rDNA locus includes multiple sequences transcribed by pol II (usually in anti-sense direction). Perhaps a very low intensity of such transcription and its sporadic character can account for the difficulty of detecting the enzyme in the nucleoli.
Best regards,
Dusan Cmarko et al.
Reviewer 2 Report
In this manuscript the authors examined if differential expression of rDNA loci is related to the nucleotide sequence of IGS in human derived HT1080 -(fibrosarcoma) cells. Authors analyzed a 2 kb long region upstream of the transcription start and have found six SNVs. They have also found a series of short deletions in a region of simple repeats correlated with the DNA methylation status. Authors observed a significant difference in repeats Q1, but not in Q2 and found three miRs matching areas that contained some of these variants. Two miRs are antisense, in the Q1 and upstream of Q2 and another one sense orientated and transcribed from the 92 bp long gene lodged within the IGS. Finally; authors suggest a correlation between the sequence variability and methylation status of the rDNA regulatory region. They highlight that single nucleotide substitutions and short deletions may play a key role in silencing ribosomal genes.
Manuscript is globally clear; however there are some points/questions should be addressed for more accuracy/comprehension. My major concern of this work is that’s mainly descriptive. Some experimental data might eventually validate (or not) the correlation proposed between the sequence variability, methylation and silencing ribosomal genes.
I also wonder how efficient is the methylated DNA immunoprecipitation reaction? And how the efficiency might affect the ratios when compared with the supernatant, considered as non-methylated DNA fraction.
Minor points
lane 173: “hsa-mir-6724 is transcribed from the 92 bp long gene lodged within IGS” Is this a PolI transcribed gene ?
lane 190: Authors should also provide more details of the deletions in Q1, since these repeats seems to be favorable for the formation of non-canonical DNA structures, include a figure for instance.
Discussion, G4 structures have been also reported in rDNA from plants by Jiri Havlová and Fajkus in 2020.
Author Response
Dear reviewer,
thank you for your comments. We revised the manuscript addressing all remarks of revisions. In particular, we provided the requested characterization of the discovered deletions.
Following are our answers:
>>>My major concern of this work is that’s mainly descriptive. Some experimental data might eventually validate (or not) the correlation proposed between the sequence variability, methylation and silencing ribosomal genes.
We agree that other experiments (focused specifically on the transcription activity of rDNA) would be needed to establish the correlation between the sequence variability and silencing of the ribosomal genes. But in the present work, as its title indicates, we were looking for a correlation between the sequence variability and methylation. Such correlation was indeed found (Table 1 and 2) for 6 SNVs and the deletions in Q1 region. Moreover, all these variants, except one SNV, proved to be related to the functional/potentially functional regions of rDNA, which allowed us to suggest certain hypothetical pathways connecting the silencing with the DNA variability. These points are explained in the 1st paragraph of the Discussion, and in its last paragraph that was accordingly modified in the revised version of our manuscript.
>>>I also wonder how efficient is the methylated DNA immunoprecipitation reaction? And how the efficiency might affect the ratios when compared with the supernatant, considered as non-methylated DNA fraction.
Since every sufficiently long fragment of DNA would contain 5-methylcytosine, we cannot expect that immunoprecipitation would separate methylated and non-methylated DNA perfectly. Nevertheless, we can, following the example of other authors (e.g. Santoro et al, 2014), use IP to assess relative abundance or enrichment of the signal. To clarify this point, we indicated that the isolated DNA “was separated into the fractions of predominantly methylated and predominantly non-methylated“ (the 1st sentence of the Results). Such separation seemed to be sufficient for our purpose, i.e. to establish the correlation between RDNA variability and its methylation..
>>>Minor points
>>>lane 173: “hsa-mir-6724 is transcribed from the 92 bp long gene lodged within IGS” Is this a PolI transcribed gene ?
The general rule seems to be that sense/antisense micro-RNAs are transcribed by pol I and pol II, respectively (Niehrs, Luke, 2020). Then hsa-mir-6724 is probably a product of pol I.
>>>lane 190: Authors should also provide more details of the deletions in Q1, since these repeats seems to be favorable for the formation of non-canonical DNA structures, include a figure for instance.
We provide further details of the deletions in Q1 and Q2 (Table 3 and the preceding paragraph of the revised version of the manuscript). Precise positions of these deletions could not be established by sequencing of the repeated regions, since identical reads may result from different deletions. Therefore, we show only the legth distribution. We also observe that this length was represented for the most part by fourfold numbers from 4 to 32. Each deletion in both Q1 and Q2 typically began and ended with the motif CTCC.
>>>Discussion, G4 structures have been also reported in rDNA from plants by Jiri Havlová and Fajkus in 2020.
We included the suggested work in the cited literature.
Best regards,
Dusan Cmarko et al.
Round 2
Reviewer 1 Report
The authors have answered satisfactorily to the comments.
Reviewer 2 Report
In the revised version of the manuscript, the authors have addressed appropriately all questions and comments . In conclusion I consider that the manuscript is now acceptable for publication.